# Taxonomic Identification of the Arctic Strain *Nocardioides Arcticus* Sp. Nov. and Global Transcriptomic Analysis in Response to Hydrogen Peroxide Stress

**DOI:** 10.3390/ijms241813943

**Published:** 2023-09-11

**Authors:** Bailin Cong, Hui Zhang, Shuang Li, Shenghao Liu, Jing Lin, Aifang Deng, Wenqi Liu, Yan Yang

**Affiliations:** 1First Institute of Oceanography, Ministry of Natural Resources, Qingdao 266061, China; shliu@fio.org.cn (S.L.); linjing@fio.org.cn (J.L.); dengaifang@fio.org.cn (A.D.); lwq07081707@163.com (W.L.); 2College of Marine Life Sciences, Ocean University of China, Qingdao 266100, China; 18829506240@163.com (H.Z.); yany@ouc.edu.cn (Y.Y.); 3College of Chemistry and Chemical Engineering, Ocean University of China, Qingdao 266100, China; shilishuanga@sina.com

**Keywords:** Arctic, *Nocardioides* sp., antioxidation, trancriptome

## Abstract

Microorganisms living in polar regions rely on specialized mechanisms to adapt to extreme environments. The study of their stress adaptation mechanisms is a hot topic in international microbiology research. In this study, a bacterial strain (Arc9.136) isolated from Arctic marine sediments was selected to implement polyphasic taxonomic identification based on factors such as genetic characteristics, physiological and biochemical properties, and chemical composition. The results showed that strain Arc9.136 is classified to the genus *Nocardioides*, for which the name *Nocardioides arcticus* sp. nov. is proposed. The ozone hole over the Arctic leads to increased ultraviolet (UV-B) radiation, and low temperatures lead to increased dissolved content in seawater. These extreme environmental conditions result in oxidative stress, inducing a strong response in microorganisms. Based on the functional classification of significantly differentially expressed genes under 1 mM H_2_O_2_ stress, we suspect that Arc9.136 may respond to oxidative stress through the following strategies: (1) efficient utilization of various carbon sources to improve carbohydrate transport and metabolism; (2) altering ion transport and metabolism by decreasing the uptake of divalent iron (to avoid the Fenton reaction) and increasing the utilization of trivalent iron (to maintain intracellular iron homeostasis); (3) increasing the level of cell replication, DNA repair, and defense functions, repairing DNA damage caused by H_2_O_2_; (4) and changing the composition of lipids in the cell membrane and reducing the sensitivity of lipid peroxidation. This study provides insights into the stress resistance mechanisms of microorganisms in extreme environments and highlights the potential for developing low-temperature active microbial resources.

## 1. Introduction

The oxygen-containing active substances formed by aerobic organisms using molecular oxygen are collectively referred to as reactive oxygen species (ROS), including superoxide anion (O^2−^), hydrogen peroxide (H_2_O_2_), hydroxyl radicals (HO ), etc., and in the process of using oxygen, the formation of ROS is inevitable [1]. Among these ROS, H_2_O_2_ is relatively stable and can gradually accumulate at lethal concentrations in the environment [2].

Transcriptomics analysis has been used to investigate the mechanisms by which strains respond to H_2_O_2_ stress. Global transcriptomic analysis showed that *Lactobacillus plantarum* CAUH2 responded to H_2_O_2_ stress by increasing carbon source utilization and enhancing glycolysis to generate more ATP, while the transcript level of antioxidant enzymes, including NADH peroxidase, thioredoxin reductase, and glutathione peroxidase, increased, and it also reduced the Fe^2+^ content to limit the effect of the Fenton response [2]. The transcriptional profiles of *Enterobacter* NRS-1 under H_2_O_2_ stress were analyzed by RNA-seq and qRT-PCR, and it was found that the differentially expressed genes (DEGs) did not include well-known major oxidative regulators, viz., *oxy*R, *sox*R, *rpo*S, *per*R, *ohr*R, so it was speculated that six factors, namely, formate dehydrogenase, processes associated with iron ions, repair programs, multidrug resistance, antioxidant defense, and energy generation (*mqo*, *sdh*C), might be involved in the response of *Enterobacter* NRS-1 to ROS stress [3]. The strategies used by different bacteria to address ROS differ, which is why we have been exploring different bacterial strategies for coping with oxidative stress.

Bacteria exist in habitats ranging from particles in clouds to deep soil, and under these different habitat conditions, bacteria must evolve different adaptation mechanisms to reduce cell damage caused by ROS [4]. The special environment of polar regions led to the bacteria inhabiting these regions evolving unique biological characteristics to adapt to the harsh environmental conditions, adopting unique survival strategies to gain survival advantages [5]. The cold subsidence of the oxygen-rich surface of the Arctic and Antarctica, coupled with the small number of organisms in deep waters, reduces the consumption of oxygen, so the oxygen content in deep seawater is relatively high. At the same time, experiments have proven that biological oxidative stress events can be induced under low-temperature conditions [6]. To survive in extreme environments, polar microorganisms have evolved their own antioxidant systems to cope with oxidative stress caused by ROS.

In this article, we describe a new bacterial species collected from the ninth Arctic scientific expedition. On the basis of the evidence presented in this study, strain Arc9.136 represents a novel species of the genus *Nocardioides*, for which the name *Nocardioides arcticus* sp. nov. is proposed. This study is a transcriptomic analysis of the response of the experimental strain Arc9.136 to H_2_O_2_ stress. Transcriptomic changes in *Nocardioides* under H_2_O_2_ stress have rarely been reported. The results of this study will enrich the resources for studying the antioxidative stress response of organisms and the existing knowledge of human diseases and microbial engineering.

## 2. Results

### 2.1. Cell Morphology and Physiology

Strain Arc9.136 was a Gram-positive, nonspore-forming and aerobic (short rods) strain. In optimal R_2_A medium, the temperature range for the growth of strain Arc9.136 was 4−37 °C, with an optimum temperature of 28 °C and no growth at 42 °C. The pH range for growth was 6–10, with an optimum of pH 7. No growth was observed at pH 5.0 or below. The NaCl range for growth was 0–7%, with an optimum of 0%. Simulated wastewater denitrification medium (KNO_3_ 0.37 g/L, sodium acetate 4.9 g/L, MgSO_4_·7H_2_O 0.126 g/L, K_2_HPO_4_ 0.49 g/L, FeSO_4_ 0.01 g/L, MnSO_4_ 0.01 g/L, filtered seawater:ultrapure water (*v/v*, 2:1)) was used to test nitrate reduction by the Arc9.136 strain, and growth was not observed. Arc9.136 tested positive for the hydrolysis of carrageenan, starch, and Tween 80 and negative for the hydrolysis of casein and algin. API 20NE, API ZYM, and Biolog tests indicated that Arc9.136 could utilize multiple carbon/nitrogen sources (Table 1).

### 2.2. The 16S rDNA Gene, ANI, and dDDH Phylogenetic Analysis

16S rDNA gene sequences of the strain Arc9.136 (1423 bp; GenBank accession number: OP861529.2) were obtained according to the method described in the Experimental Procedures section. Phylogenetic analysis based on the 16S rDNA gene sequence indicated that strain Arc9.136 belonged to the genus *Nocardioides* and shared the highest 16S rDNA gene sequence similarity with *Nocardioides marmotae* zg-579 (97.82%), followed by *Nocardioides pacificus* XH274 (96.69%), *Nocardioides lianchengensis* D94-1^T^ (96.56%), *Nocardioides marinisabuli* SBS-12^T^ (96.69%), and *Nocardioides deserti* SC8A-24^T^ (96.42%). The Arc9.136 strain was shown to be related to *Nocardioides,* and the precise evolutionary relationship was discovered by phylogenetic analysis (Figure 1).

The results of the average nucleotide identity (ANI) analysis of strain Arc9.136 and the four reference strains *N. marmotae*, *N. lianchengensis*, *N. deserti*, and *N. marinisabuli* are shown in Figure 2B. Arc9.136 had the highest ANI comparison rate with *N. marmotae* at 86.14%, followed by *N. deserti* at 85.89%, *N. lianchengensis* at 79.13%, and the lowest ANI comparison rate with *N. marinisabuli* at 78.65%, all less than 95% (Table 2). At the same time, all the dDDH values were less than 70% (Table 2).

### 2.3. Whole-Genome Assembly and Annotation

The complete genome of Arc9.136 had one 4,414,287 bp circular chromosome with a G + C% content of 73.61 mol% (Figure 2A). Bacterial coding gene prediction was performed using GeneMarkS software (verison 4.17), and 4249 genes were predicted. The genome also contained 3 5S rRNA, 3 16S rRNA, 3 23S rRNA and 45 tRNA, respectively. Six gene islands were found (Table 3). The whole genome sequences of Arc9.136 were aligned with other related sequences (https://tygs.dsmz.de/, accessed on 23 August 2023) (Appendix A).

The amino acid sequences of the target species were compared with the GO database using Interproscan software (verison 4.8), and the genes of the target species were combined with their corresponding functional annotation information to obtain the annotation results.

Arc9.136 was annotated through the GO database, and a total of 10,577 genes were annotated. The number of genes related to biological processes was 5365, accounting for 50.72% of the total number of genes; 1803 genes were related to cellular components, accounting for 17.05% of the total genes; and 3409 genes were related to molecular functions, accounting for 32.23% of the total genes. In the biological process category, the number of genes related to metabolic processes was the largest, at 1528. In the cellular component category, the number of genes related to cell parts was the largest, at 716. In the molecular function category, the number of genes related to catalytic activity was the largest, at 1584 (Figure 2B). In the molecular function category, the number of genes related to antioxidant activity was nine (Appendix A).

### 2.4. Effect of H_2_O_2_ Concentrations on the Growth of Arc9.136

The sensitivity of Arc9.136 to H_2_O_2_ was tested, and significant bacterial growth was observed at H_2_O_2_ concentrations of 0.5, 1, and 2 mM, while no significant bacterial growth was observed at other concentrations. Our results show that the MIC of H_2_O_2_ for Arc9.136 was 2 mM. R_2_A agar plates were inoculated with 2 μL of the bacterial suspension taken from R_2_A medium in which there was no microbial growth, and we concluded that the MBC of H_2_O_2_ for Arc9.136 was 15 mM. Therefore, Arc9.136 was treated with 1 mM H_2_O_2_ to further investigate its transcriptional response under oxidative stress.

### 2.5. Overview of the Arc9.136 Transcriptomic Response to H_2_O_2_

The transcriptome-level changes in Arc9.136 strains treated with H_2_O_2_ (H-1, H-2, H-3) and untreated strains (1-1, 1-2, 1-3) are shown in Table 4. As shown in Figure 3A, three independent biological replicates in groups A and B were clustered together. Differential expression analysis showed that the 268 significant DEGs consisted of 169 upregulated and 99 downregulated genes, as shown in Figure 3B (*p* value ≤ 0.05, |log_2_FoldChange| ≥ 0.5).

All 268 DEGs were annotated against the COG database to further explore the functions of the DEGs and their metabolic pathways. The putative functions of these genes were classified into different categories grouped by COG (Figure 3C,D). The DEGs were involved in various biological processes, such as carbohydrate transport and metabolism, inorganic ion transport and metabolism, defense mechanisms, replication, recombination and repair, and cell wall/membrane/envelope biogenesis, which are discussed in detail as follows.

### 2.6. Real-Time Quantitative PCR Analysis

To verify the reliability of the transcriptomic data, six genes were selected for qRT-PCR validation. The experimental results showed that the genes were upregulated and expressed, which was consistent with the transcriptomic results, indicating that the transcriptomic data were credible (Figure 4).

## 3. Discussion

In this paper, the strain Arc9.136 collected during the ninth Arctic scientific expedition was identified by multiphase classification, and this strain was found to grow well in R_2_A broth supplemented with hydrogen peroxide. This suggests that the Arc9.136 strain collected from polar habitats has antioxidant potential. We also investigated the Arc9.136 antioxidant mechanism through genome-wide and transcriptome analyses.

In experiments, we found that Arc9.136 grew best in R_2_A broth. The strain showed optimal growth at 28 °C in R_2_A broth with an optimal salinity of 0%; these conditions, surprisingly, were different from the optimal temperature and optimal salinity of other deep-sea bacteria, as initially suspected. At the same time, Arc9.136 cells appeared as short rods. The API 20NE Reagent Strip is made up of 20 tubes containing a dry substrate. These tubes are used by inoculating with a bacterial suspension and culturing for a certain period, and the results are observed as metabolism-based color changes. The results showed that Arc9.136 had β-D-glucosidase and β-galactosidase activities and could assimilate maltose and glucose. We also used API ZYM to detect enzyme activity and used the Biolog system to test the biochemical reactions of sole carbon sources. We also found that Arc9.136 had the ability to hydrolyze starch and carrageenan (Table 1).

In bacterial genomes, the gene encoding 16S rDNA exhibits good evolutionary conservation, a suitable analysis length, and good variability matching the evolutionary distance, and it has become a standard marker sequence for bacterial molecular identification [7]. BLASTN using the 16S rDNA gene showed that Arc9.136 shared the highest similarity with *N. marmotae* (97.82%), at less than 98%. 16S rDNA gene sequencing and phylogenetic analysis showed that Arc9.136 was a novel species of the genus *Nocardioides*. The bacteria were also subjected to ANI analysis; the ANI is defined as the average base similarity between homologous segments of two microbial genomes and has great benefits in distinguishing closely related species. Compared with DNA–DNA hybridization (DDH), ANI analysis calculation is simple and quick. We downloaded the whole-genome sequences of *N. marmotae, N. lianchengensis*, *N. deserti*, and *N. marinisabuli* from the NCBI database and compared them using OrthoANI. The results showed that the Arc9.136 strain shared 78.65%−86.14% similarity, all less than 95%, when the ANI ≥ 95%, and it seems that we can assume that they belong to the same species [8]. We also found that the major fatty acids of Arc9.136 were iso-C_16:0_ (10.66%), iso-C_17:1_ ω9c (14.58%), and iso-C_17:0_ (16.77%) (Table 1). The predominant menaquinone is MK-8(H_4_). The polar lipids consisted of phosphatidyl ethanolamine, phosphatidyl glycerol, amino phospholipid, phosphatidyl inositol, and an unidentified lipid. Thus, it was further proven that Arc9.136 is a novel species of the genus *Nocardioides*.

Illumina and PacBio sequencing technologies are now widely used to assemble genomes because of their low cost and high efficiency [9,10]. As a third-generation sequencing technology, PacBio sequencing is relatively expensive, but the sequencing reads are longer, while Illumina platform data are stable and reliable, highly reproducible, high quality, and low cost; usually, a combination of sequencing data from the two platforms is used. In this study, we used PacBio sequencing data to assemble the strain chromosome sequence and used Illumina sequencing data to correct the error. We found that Arc9.136 had one 4,414,287 bp circular chromosome and a DNA C + G mol% content of 73.61 mol% (Table 2). The molar percentage of G + C mol% in strain DNA was between 66.5 and 78.7 mol% [11].

*Nocardioides* was described as a genus of Gram-positive bacteria with a high G + C content (68.7–74.9 mol%) that are spherical or rod-shaped [11]. The main features of this genus include iso-C_16:0_ as the major fatty acid and MK-8 (H_4_) as the main menaquinone [12,13]. Through 16S rDNA gene sequencing, ANI analysis, and determination of the levels of major fatty acids and the G + C mol%, we finally confirmed that strain Arc9.136, isolated during the ninth Arctic scientific expedition, is clearly different from all other recognized *Nocardioides* strains. We propose the name *Nocardioides arcticus* sp. nov. to designate the Arc9.136 strain.

The antioxidant mechanism of Arc9.136 is certainly related to its genetic material, so analysis of its genetic information would help to reveal the antioxidant mechanism. A total of nine genes in the GO database were annotated with antioxidant activity (Appendix A). The gene Arc9.136_GM003016 encodes a superoxide dismutase (SOD). SODs can catalyze superoxide disproportionation reactions, scavenging O^2−^, resulting in the formation of H_2_O_2_ [14,15]. This may explain why the gene Arc9.136_GM003016 was not a DEG. The genes Arc9.136_GM000075, Arc9.136_GM000051, Arc9.136_GM000247, and Arc9.136_GM001188 encode glutathione peroxidases. Glutathione peroxidases catalyze the reduction of H_2_O_2_ by oxidizing glutathione [16]. *L. plantarum* HFY09, by upregulating the expression of glutathione peroxidase, defends cells against oxidative damage from alcohol [17]. Some antioxidant enzymes, including glutathione peroxidase, were upregulated 6.23-fold at the transcript level for H_2_O_2_ scavenging in *Lactobacillus plantarum* CAUH2 [2]. The genes Arc9.136_GM000471 and Arc9.136_GM001349 encode peroxidases. The genes Arc9.136_GM000758 and Arc9.136_GM002112 encode catalases. Hydrogen peroxide is usually scavenged by peroxidase or catalase [18]. The formation of ROS in *Corynebacterium glutamicum* is induced by acid stress, and peroxidase protects *C. glutamicum* against acid stress through the reduction of intracellular ROS levels [19]. Catalases convert hydrogen peroxide to oxygen and water (2H_2_O_2_ → 2H_2_O + O_2_) [4]. These genes could play a role in Arc9.136 antioxidant behavior. These results were unexpected; the sequencing results were the opposite of the expected results, and none of the genes was significantly differentially expressed. This suggests that Arc9.136 differs from conventional strains in its response to oxidative stress, which may occur via a novel antioxidant mechanism that needs to be further confirmed.

In this study, we performed transcriptomic analysis of two groups of samples, 1 mM H_2_O_2_-treated (H-1, H-2, H-3) and nontreated (1-1, 1-2, 1-3) samples. The results showed that in Arc9.136 under 1 mM H_2_O_2_ stress, a total of 268 genes were significantly differentially expressed, of which 169 genes were upregulated and 99 genes were downregulated. We mainly analyzed the upregulated genes annotated against the COG database (Figure 3).

A total of 22 genes were annotated in the carbohydrate transport and metabolism functional category (Table 4). Almost all genes involved in carbohydrate transport and metabolism were upregulated in response to H_2_O_2_ stress. Among them, the genes Arc9.136_GM000490, Arc9.136_GM000489, and Arc9.136_GM000488 encoding a D-xylose transport system permease protein, D-xylose transport system ATP-binding protein, and D-xylose transport system substrate-binding protein were upregulated 2.12-, 2.63-, and 2.17-fold, respectively. The gene Arc9.136_GM000764, encoding a sugar phosphate isomerase, was upregulated 3.35-fold. Most sugar phosphate isomerases are present in a variety of microorganisms because they participate in the pentose phosphate pathway and glycolytic metabolism and at the same time have wide-ranging substrate specificity, exhibiting great potential in the enzymatic production of various rare sugars [20]. The gene Arc9.136_GM003971, encoding a beta-glucosidase, was upregulated 2.39-fold. Beta-glucosidase is an integral part of the cellulase enzyme complex and plays an extremely important role in the complete hydrolysis of cellulose to glucose. Moreover, the genes Arc9.136_GM001007, Arc9.136_GM001008, Arc9.136_GM000765, and Arc9.136_GM000767, encoding the ribose/D-xylose ABC transport system, were upregulated 2.49-, 2.14-, 1.83-, and 1.77-fold, respectively. The genes Arc9.136_GM003973, Arc9.136_GM003974, and Arc9.136_GM003975, encoding the raffinose/stachyose/melibiose ABC transport system, were upregulated 1.84-, 1.67-, and 1.79-fold, respectively. At the same time, the genes Arc9.136_GM000966, Arc9.136_GM000967, Arc9.136_GM000968, and Arc9.136_GM000969, encoding multiple sugar ABC transport systems, were upregulated 1.65-, 1.62-, 1.63-, and 1.62-fold, respectively. Therefore, to adapt to the oxidative stimulation caused by H_2_O_2_, Arc9.136 makes efficient use of various carbon sources, allowing it to cope with oxidative stress.

There were also 18 genes annotated to the inorganic ion transport and metabolism functional category (Table 4). The genes encoding an iron complex transport system substrate-binding protein (Arc9.136_GM002551), the iron uptake system component EfeO (Arc9.136_GM001348), a ferric iron ABC transporter (Arc9.136_GM001365), a ferric iron ABC transporter permease (Arc9.136_GM001364), and an ABC transporter ATP-binding protein (Arc9.136_GM002459) were upregulated 1.58-, 1.62-, 1.56-, 1.68-, and 1.53-fold, respectively. Notably, no gene encoding a ferrous iron-related protein was downregulated. At the same time, genes associated with sulfur metabolism were downregulated. The gene Arc9.136_GM002509, encoding a sulfite reductase, was downregulated 0.624-fold. The gene Arc9.136_GM000630, encoding a sulfate ABC transporter ATP-binding protein, was downregulated 0.560-fold. Iron is necessary for almost all living organisms because many enzymes use iron as a cofactor [21]. Under ROS stress, the use of ferrous iron is inhibited to prevent the Fenton reaction and reduce the production of HO·, but at the same time, ferric iron becomes the main source of intracellular homeostasis [2]. Excessive iron is toxic to cells because it catalyzes the formation of ROS [21], but almost all organisms require iron to participate in basic cellular processes and synthesize high-energy compounds to provide fuel for cell growth and proliferation [22]. However, to limit the Fenton reaction to reduce the production of HO·, cells must reduce the level of Te^2+^. The species *Streptococcus thermophilus* responds to H_2_O_2_ stress by reducing the intracellular iron concentration [23]. Therefore, to avoid irreparable damage caused by HO· to cells, the Arc9.136 strain must reduce the intracellular Te^2+^ content and maintain intracellular iron homeostasis by increasing the content of Te^3+^. However, we found no downregulation of Te^2+^-related genes by transcriptomic analysis.

In addition, seven genes were annotated to the defense mechanisms, replication, recombination, and repair functional category (Table 4). The gene Arc9.136_GM002892 (dnaE2), encoding an error-prone DNA polymerase, was upregulated 4.37-fold. The major role of error-prone polymerases is to provide a means to address environmental damage to the genome [24]. The gene Arc9.136_GM002764 (DnaB), encoding a replicative DNA helicase, was upregulated 2.80-fold. DnaB is the main replicative helicase of *E. coli* and migrates along the lagging actively during the progress, serving both to unwind the DNA duplex in advance of the leading strand and to potentiate synthesis by the bacterial primase of RNA primers for the nascent (Okazaki) fragments of the lagging strand [25]. The gene Arc9.136_GM002761 (DNMT1), encoding a DNA cytosine methyltransferase, was upregulated 2.76-fold. DNMT1 is a DNA methyltransferase that functions during DNA replication to methylate hemimethylated sites; that is, DNMT1 methylates cytosines in CpG dinucleotides on newly replicated strands in the wake of the replication fork through methylated DNA regions. DNMT1 is important at the epigenetic level for maintaining DNA methylation in cells, and it is precisely because DNMT1 retains methylation that DNA replication does not lose this function. The gene Arc9.136_GM001599 (dinB), encoding DNA polymerase IV, was upregulated 3.66-fold. DNA polymerases are a group of enzymes that are used to make copies of DNA templates and are essentially used in DNA replication mechanisms. These enzymes make new copies of DNA from existing templates and function by repairing the synthesized DNA to prevent mutations. DNA polymerase IV can play a role in the SOS response and is involved in DNA repair and damage tolerance pathways [26]. Therefore, we speculate that Arc9.136 uses four genes, dnaE2, DnaB, DNMT1, and dinB, to repair DNA damage caused by H_2_O_2_.

We also observed that the genes Arc9.136_GM001729 (ABC-2.A) and Arc9.136_GM001730 (ABC-2.P) were upregulated 1.83-fold and 1.84-fold, respectively. These two genes encode an ABC-2 type transport system ATP-binding protein and ABC-2 type transport system permease protein, both of which belong to the ABC type multidrug transport system. We speculate that this may be related to the oxidative stress response (Table 5).

An additional 55 genes were annotated to the energy production and conversion and lipid transport and metabolism functional categories (Appendix A). Changes in plasma membrane components may also be a strategy in the response to oxidative stress [27,28].

## 4. Experimental Procedures

### 4.1. Bacterial Strains and Growth Conditions

The strain Arc9.136 was isolated from the marine sediments (84°34′54″N, 162°09′31″W, 2178.3 m) collected during the ninth Arctic scientific expedition and kept at −80 °C, deposited in the China Center for Type Culture Collection Center (CCTCC) under accession number CCTCC AA 2023014. Before use, the bacteria were first revived from a −80 °C glycerol stock. Next, the revived strains were inoculated (1%) on four media, namely, ISP-2 medium, R_2_A medium, glucose asparagine medium, and Zobell 2216E medium (Appendix A). All media were heat-sterilized at 121 °C for 20 min.

### 4.2. Characterization of Arc9.136

Arc9.136 was isolated, and its morphology was described using a scanning electron microscope (SEM) (Figure 5). Cells collected during the exponential and stationary growth phases were used for Gram staining [29]. The temperature range for growth was determined by incubation of the strain in R_2_A broth at 4, 16, 28, 37, and 42 °C [30]. The pH range for growth was measured in R_2_A broth at pH 4.0–12.0 (at intervals of 1 pH unit) [31]. Salt tolerance was tested in R_2_A broth without added NaCl or supplemented with 1, 3, 5, 7, and 10% (*w/v*) NaCl [32]. The hydrolysis of starch, casein, carrageenan, alginase, and Tween 80 was examined as previously described [33]. API 20NE (BioMerieux, France), API ZYM, and Biolog GEN III MicroPlates were used according to the manufacturer’s instructions. The strain Arc9.136 was cultured in R_2_A broth under optimum conditions. Strains in the mid-exponential growth phase were collected and subjected to freeze-drying treatment. The polar lipids were extracted with a solution consisting of chloroform, methanol, and water (65:25:4, *v/v/v*) and were analyzed by two-dimensional TLC as described previously [34]. Detection and analysis of fatty acid composition and content were carried out using gas chromatography (Agilent 7890A) [35]. High-performance liquid chromatography was used to determine the content and types of respiratory menaquinone [36].

### 4.3. 16S rDNA Extraction, Molecular Determination, and Phylogenetic Analysis

The DNA of the Gram-positive Arc9.136 strain was extracted using the TIANamp Bacteria DNA Kit (Tiangen Biotech, Beijing). After DNA extraction and purification, electrophoresis on a 1% agarose gel was used to determine DNA quality. To amplify the 16S rDNA gene, we used the primers 27F (5′-AGAGTTTGATCCTGGCTCAG-3′) [37] and 1492R (5′-GGTTACCTTGTTACGACTT-3′) [38] under the following reaction conditions: 94 °C for 5 min; 30 cycles of 94 °C for 1 min, 55 °C for 30 s, and 72 °C for 90 s; 72 °C for 10 min. The PCR product was sent to Sangon Biotech (Shanghai, China) Co., Ltd. for sequencing, and the obtained sequences were compared by BLASTN analysis against the NCBI database [39] to identify its closest relatives (https://blast.ncbi.nlm.nih.gov/Blast.cgi, accessed on 30 November 2022). A phylogenetic tree was established based on a representative set of related 16S rDNA gene sequences from the GenBank database [40] by using the neighbor-joining method and bootstrap analysis using 1000 resampled datasets in MEGA 6.0 software(version 6.0) [41,42]. OrthoANI (version 0.93.10) was used to evaluate the ANI values between the genome sequence of strain Arc9.136 and those of *N. marmotae*, *N. deserti*, *N. lianchengensis*, and *N. marinisabuli* [43]. The genomic distances and digital DNA–DNA hybridization (dDDH) were calculated by using the Genome-to-Genome Distance Calculator (https://tygs.dsmz.de/, accessed on 23 August 2023) [44].

### 4.4. Whole-Genome Sequencing and Assembly

The Arc9.136 strain was cultivated at 28 °C and 150 r/min to exponential phase and harvested by centrifugation at 4 °C (12,000× *g*, 20 min). The genome of Arc9.136 was sequenced by single-molecule, real-time (SMRT) technology. Sequencing was performed at Novogene Bioinformatics Technology Co., Ltd. (Beijing, China) by using a PacBio RS II sequencer (Pacific Biosystems, Menlo Park, CA, USA) [45,46,47].

The genomic DNA of the Arc9.136 strain was extracted using the SDS method according to Xia et al. [48]. The SMRT Bell TM Template kit (version 1.0) was used to build 10K SMRT Bell libraries, and quantification was performed by a Qubit. NEBNext^®^Ultra™ DNA Library Prep Kit for Illumina (NEB, Ipswich, MA, USA) was used to build a 350 bp small-fragment library, and quantification was performed by a Qubit. Then, sequencing of two libraries was carried out using PacBio RS II SMRT and Illumina NovaSeq PE150 sequencing technologies (Beijing NuoheZhiyuan Technology Co., Ltd., Beijing, China).

Clean reads were assembled using SMRT Link v5.1.0 software (version 5.1.0) [49,50]. Arrow software(version 2.2.1) was used to optimize assembly results. Encoding gene predictions were performed using GeneMarkS software(verison 4.17) [51]. Sequencing of tRNA and rRNA was performed using tRNAscan-SE (version 1.3.1) and rRNAmmer software (version 1.2), respectively [52,53]. Finally, the gene functions were further investigated by BLASTP using the following five different function databases: the nonredundant (NR) protein database (http://www.ncbi.nlm.nih.gov/protein, accessed on 30 November 2022) [54], the clusters of orthologous groups (COG) database (https://www.ncbi.nlm.nih.gov/COG/, accessed on 30 November 2022)) [55], the GO database (http://www.geneontology.org/, accessed on 30 November 2022) [56], and the KEGG database (http://www.genome.jp/kegg/pathway.html, accessed on 30 November 2022) [57]. Moreover, a more specific and detailed analysis of the presence of gene clusters related to antioxidant substance production and other secondary metabolites was performed using antiSMASH (version 5.1). The G + C mol% of the DNA was determined from the whole-genome sequence.

### 4.5. Physiological Characterization of the Response of Arc9.136 to H_2_O_2_

The response of the Arc9.136 strain to H_2_O_2_ was assessed by a variety of methods. Broth dilution and disc diffusion assays were employed to determine the minimum inhibitory concentration (MIC) and minimum bactericidal concentration (MBC), respectively. The MIC was defined as the lowest H_2_O_2_ concentration at which no growth of microorganisms was observed. The MBC was defined as the lowest concentration of H_2_O_2_ that completely eliminated the viability of the tested bacteria [58]. First, Arc9.136 in the mid-log phase (OD_600_ ≈ 0.6) was inoculated into 5 mL of R_2_A medium at an inoculum size of 1% and incubated for 24 h at 28 °C; H_2_O_2_ was added at different concentrations (1, 2, 3, 4, 5, 6, 7, 8, 9,10, 15, 20, 25, 30, 35 mM) to the medium. In the next step, the R_2_A agar plates were inoculated with 2 μL of a bacterial suspension taken from the R_2_A medium in which no microbial growth was observed. All experiments were repeated at least three times.

### 4.6. Transcriptomic Analysis

For transcriptomic analysis, cell samples were prepared as described previously [59]. First, the Arc9.136 strain in the mid-log phase (OD_600_ ≈ 0.6) was cultured at 28 °C and 150 r/min for 1.5 h under 1 mM H_2_O_2_ stress. Then, the control and sample cells were pelleted at 8000 rpm for 30 min at room temperature and frozen immediately in liquid nitrogen. Triplicate samples were prepared for each condition. Total RNA was extracted with TRIzol reagent, and the RNA concentration was measured using a Qubit RNA Assay Kit in a Qubit 2.0 fluorometer (Shenzhen, Shanghai). RNA integrity and genomic contamination were detected on agarose gel.

### 4.7. qRT-PCR Analysis

qRT-PCR analysis was performed to validate the RNA-seq transcriptome profiling experiments. According to the genome sequence of the strain Arc9.136 and related references [60], glyceraldehyde 3-phosphate dehydrogenase (GAPDH) was selected as the candidate reference gene for this experiment. At the same time, significant DEGs (Appendix A) were selected for qRT-PCR to verify the reliability of the transcriptomic data. Relative gene expression was calculated by the 2^−ΔΔCt^ method.

## 5. Conclusions

Our results describe a new strain of *Nocardioides arcticus* (arc’ti.cus. L. masc. adj. *arcticus*, northern, pertaining to the Arctic), with a circular chromosome with a total length of 4,414,287 bp, containing 4239 coding genes, of which 9 genes are annotated with antioxidant activity. Experiments showed that strain Arc9.136 may respond to oxidative stress through the following strategies:(1)Improving carbohydrate transport and metabolism and efficiently utilizing various carbon sources give strain Arc9.136 a strong survival advantage, while producing relatively high amounts of ATP for DNA and protein repair and metal ion transport.(2)By altering inorganic ion transport and metabolism, reducing divalent iron uptake and increasing the use of ferric iron, the Fenton reaction is prevented, thereby reducing the damage of oxidative stress to cells while maintaining intracellular iron homeostasis.(3)Arc9.136 can enhance cell replication, DNA repair, and defense functions, reducing H_2_O_2_-mediated damage in cells and improving its survival rate.(4)Arc9.136 can alter the fluidity of its lipid membrane by changing the composition of lipids in the cell membrane, thereby affecting the transduction of environmental stress signals and reducing the sensitivity of cell membranes to lipid peroxidation.

Since there are few studies on the antioxidant effects of *Nocardioides*, several scientific questions remain unanswered. For example, what are the specific adaptation mechanisms of *Nocardioides* to oxidative stress? Does the new *Nocardioides* species discovered in this study have a new antioxidant mechanism?

We will further analyze the key genes suitable for hydrogen peroxide stress, explore new key functional genes, continue to study the natural products of Arc9.136, reveal its antioxidant function at the protein level, and further clarify its application potential and antioxidant mechanism.

## Figures and Tables

**Figure 1 ijms-24-13943-f001:**
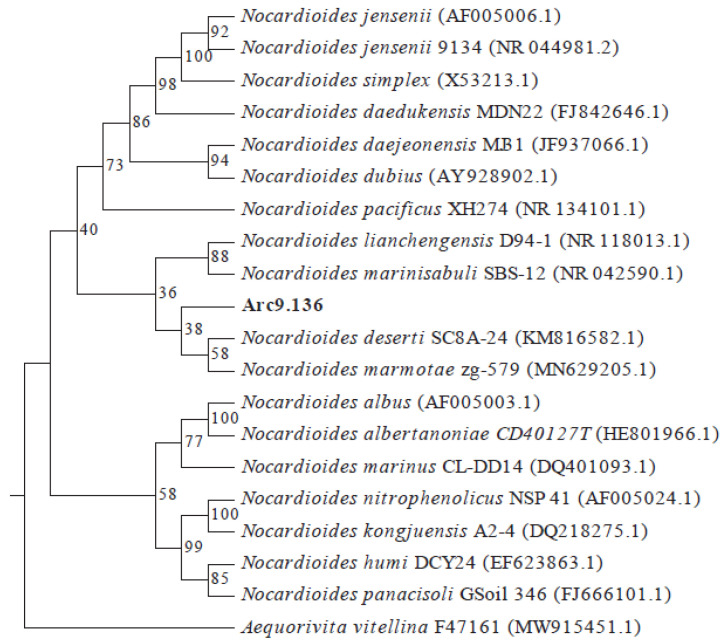
Phylogenetic relationship of strain Arc9.136. Neighbor-joining phylogenetic tree showing the position of the isolate Arc9.136 within the genus *Nocardioides* based on 16S rDNA gene sequence data. Bootstrap ≥ 1000 (accession number: OP861529.2).

**Figure 2 ijms-24-13943-f002:**
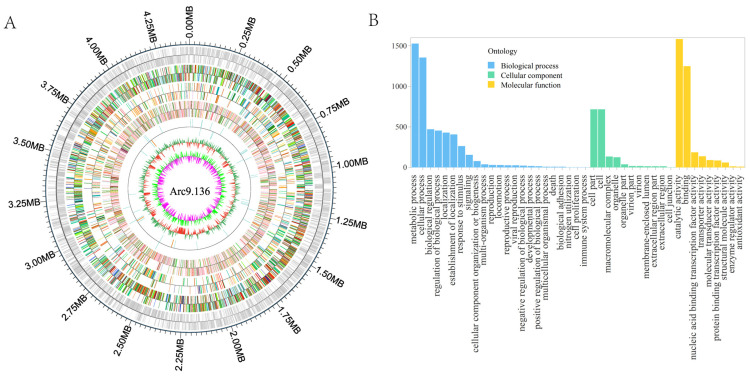
Whole-genome assembly and annotation of Arc9.136. (**A**) Circular map of the Arc9.136 genome. From inside to outside, the first innermost circle represents genome GC skew values, the second circle describes genome GC content, the third circle shows ncRNA, the fourth, fifth, and sixth circles are GO, KEGG, and COG function annotations, respectively. The seventh circle is genes. (**B**) GO analysis of Arc9.136.

**Figure 3 ijms-24-13943-f003:**
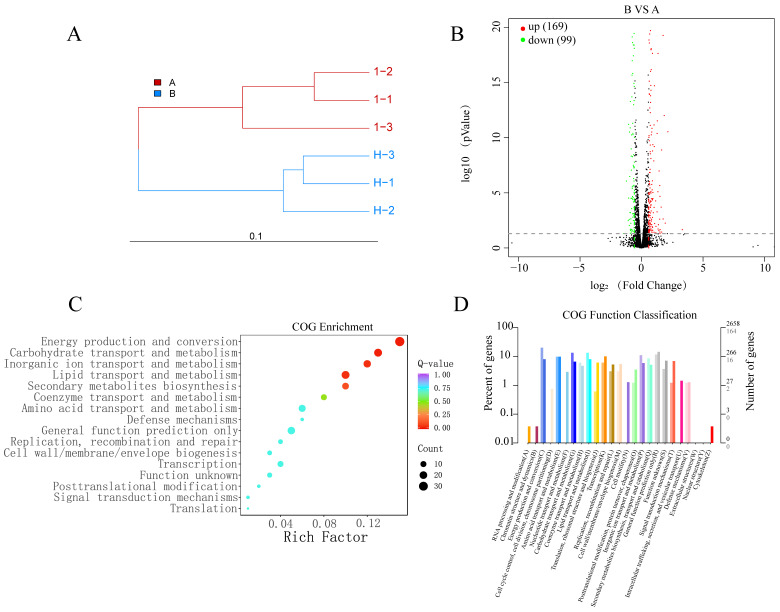
Transcriptome sequencing of Arc9.136 under 1 mM H_2_O_2_. (**A**) Cluster analysis of the H_2_O_2_-treated group (group B: H-1, H-2, H-3) and nontreated group (group A: 1-1, 1-2, 1-3). The length of the branch represents the distance between samples. (**B**) Volcano plot showing the DEGs between the H_2_O_2_-treated group (group B: H-1, H-2, H-3) and nontreated group (group A: 1-1, 1-2, 1-3). The X-axis indicates the fold change in gene expression (threshold, |log2 (treatment/control)| ≥ 0.5), while the Y-axis indicates the statistically significant level (threshold, *p* value ≤ 0.05). (**C**) Function enrichment scatter plot of DEGs by COG annotation. The rich factor represents the ratio of the number of target genes to the total number of annotated genes in this pathway. (**D**) Histogram analysis of DEGs by COG annotation.

**Figure 4 ijms-24-13943-f004:**
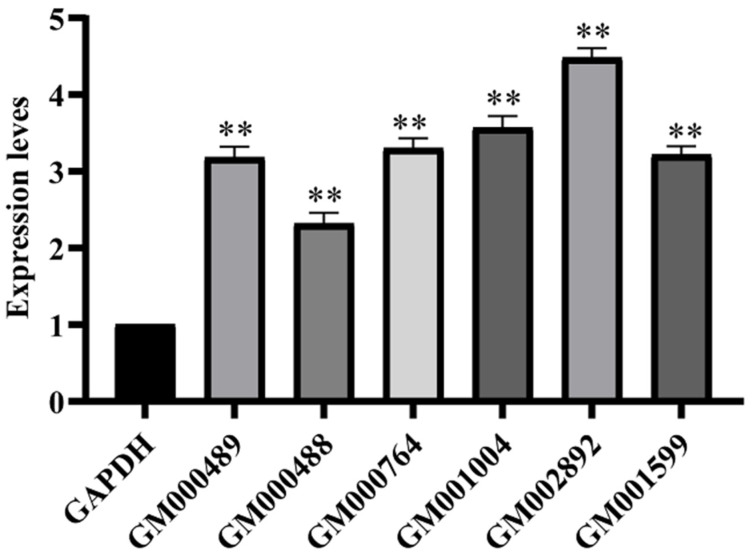
Verification of gene expression levels by real-time qRT-PCR analysis. Significant difference indicated by ** *p* < 0.01.

**Figure 5 ijms-24-13943-f005:**
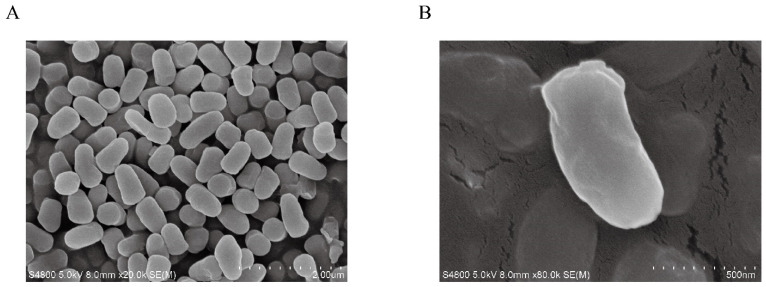
Scanning electron microscopy (SEM) images of Arc9.136. Scale (**A**) 200 μm; Scale (**B**) 500 nm.

**Table 1 ijms-24-13943-t001:** Phenotypic characteristics differentiating strain Arc9.136 from related species of the genus *Nocardioides*. All strains were Gram-positive, positive for carrageenan, and negative for nitrate reduction and alginate. Arc9.136 was positive for dextrin, D-maltose, D-trehalose, D-cellobiose, D-raffinose, D-melibiose, β-methyl-D-glucoside, α-D-glucose, D-mannose, D-fructose, D-galactose, D-sorbitol, D-mannitol, D-arabitol, myo-inositol, gelatin, L-aspartic acid, pectin, D-galacturonic acid, D-gluconic acid, D-glucuronamide, D-lactic acid methyl ester, L-lactic acid, γ-amino-butyric acid, β-hydroxy-D, L-butyric acid, α-keto-butyric acid, acetoacetic acid, propionic acid, acetic acid, D-serine, nalidixic acid, potassium tellurite, and aztreonam and negative for others.

Characteristics	Arc9.136	*N. deserti*	*N. lianchengensis*	*N. marinisabuli*
Cell morphology	Short rods	Cocci	Rods	Rods
Cell size (μm)	0.2–0.4 × 0.4–0.6	0.3–0.6 × 0.3–1.1	0.4 × 1.1–2.5	0.6–0.8 × 1.4–2.1
pH range (optimum) for growth	6–10 (7)	5.0–12.0 (7)	6–9 (7)	6–12 (7)
Temperature range (optimum) for growth (°C)	4–37 (28)	10–42 (30)	10–40 (30)	4–40 (30)
NaCl range (optimum) for growth (%, *w/v*)	0–7 (0)	0–7 (0)	0–4 (0–2)	0–8 (0)
Nitrate reduction	–	–	–	–
Hydrolysis of				
Starch	+	+	+	–
Tween 80	+	+	+	–
Carrageenan	+	+	+	+
Casein	–	–	+	–
Alginate	–	–	–	–
Enzyme activities (API ZYM)				
Alkaline phosphatase	–	+	–	+
Esterase (C4)	+	+	–	–
Esterase lipase (C8)	+	–	–	+
Leucine arylamidase	+	–	+	–
Valine arylamidase	+	–	+	–
Cystine arylamidase	+	+	–	–
Trypsin		+	–	–
Acid phospholipase	–	–	±	–
Naphthol-AS-BI-phosphohydrolase	+	–	+	–
α-Glucosidase	+	+	+	+
*β*-Glucosidase	+	+	+	–
DNA G + C content (mol%)	73.61	71.00	71.80	73.10
Fatty acids (>10% of total fatty acids) (%)				
	iso-C_16:0_ (10.66)	iso-C_16:0_ (8.94)	iso-C_16:0_ (29.15)	iso-C_16:0_ (48.70)
	iso-C_17:1_ ω9c (14.58)	C_17:1_ω8c (12.63)	anteiso-C_17:0_ (21.00)	
	iso-C_17:0_ (16.77)	10-Me C_17:0_ (11.58)		
		C_18:1_ω9c (10.28)		

Symbols: +, positive; –, negative.

**Table 2 ijms-24-13943-t002:** Most closely related *Nocardioides* species as results from whole-genome comparisons.

Arc9.136 Genome-to-Genome Comparisons				
	*N. marmotae* (GCF_013177455.1)	*N. deserti* (GCF_014646035.1)	*N. lianchengensis* (GCF_900101465.1)	*N. marinisabuli* (GCF_013466785.1)
ANI (in %)	86.14	85.89	79.13	78.65
dDDH (d4, in %)	29.80	29.60	21.90	21.50

**Table 3 ijms-24-13943-t003:** Genome characteristics of Arc9.136.

Item	Description
Size (bp)	4,414,287
G + C content (%)	73.61
Gene islands	6
tRNA	45
5S rRNA	3
16S rDNA	3
23S rRNA	3
CDs	4249

**Table 4 ijms-24-13943-t004:** Summary of RNA-seq data for the control (1-1, 1-2, 1-3) and H_2_O_2_ treatment groups (H-1, H-2, H-3) of strain Arc9.136.

Sample	1-1	1-2	1-3	H-1	H-2	H-3
Read length (bp)	139.48	138.44	138.45	136.42	137.43	135.29
Raw reads	31,611,962	36,563,802	31,064,278	37,358,194	34,224,468	35,946,166
Clean reads	30,880,670	35,632,360	30,271,138	36,454,510	33,222,756	34,950,364
Clean bases (bp)	4,307,162,951	4,932,987,542	4,191,127,043	4,972,957,368	4,565,921,804	4,728,589,787
Q20 (%)	98.57%	98.51%	98.50%	98.54%	98.40%	98.42%
Q30 (%)	95.05%	94.86%	94.82%	95.00%	94.59%	94.69%
Mapped (%)	99.57%	99.53%	99.53%	99.46%	99.49%	99.41%

**Table 5 ijms-24-13943-t005:** Gene expression patterns at the transcriptional level in Arc9.136 with 1 mM H_2_O_2_.

Gene_ID	Gene	Fold Change	Description
Carbohydrate transport and metabolism			
Arc9.136_GM000490	xylH	2.12	D-xylose transport system permease protein
Arc9.136_GM000489	xylG	2.63	D-xylose transport system ATP-binding protein
Arc9.136_GM000488	xylF	2.17	D-xylose transport system substrate-binding protein
Arc9.136_GM000764	dpe	3.35	sugar phosphate isomerase
Arc9.136_GM003971		2.39	beta-glucosidase
Arc9.136_GM001007	rbsA	2.49	ribose transport system ATP-binding protein
Arc9.136_GM001008	rbsC	2.14	ribose transport system permease protein
Arc9.136_GM000765	rbsB	1.83	ribose transport system substrate-binding protein
Arc9.136_GM000767	rbsC	1.77	ribose transport system permease protein
Arc9.136_GM003973	msmG	1.84	raffinose/stachyose/melibiose transport system permease protein
Arc9.136_GM003974	msmF	1.67	raffinose/stachyose/melibiose transport system permease protein
Arc9.136_GM003975	msmE	1.79	raffinose/stachyose/melibiose transport system substrate-binding protein
Arc9.136_GM001006		2.97	sugar ABC transporter substrate-binding protein
Arc9.136_GM000967	ABC.MS.P1	1.62	multiple sugar transport system permease protein
Arc9.136_GM000968	ABC.MS.P	1.63	multiple sugar transport system permease protein
Arc9.136_GM000969	ABC.MS.S	1.62	multiple sugar transport system substrate-binding protein
Inorganic ion transport and metabolism			
Arc9.136_GM002551		1.58	iron complex transport system substrate-binding protein
Arc9.136_GM001348	efeO	1.62	iron uptake system component EfeO
Arc9.136_GM001365		1.56	ferric iron ABC transporter
Arc9.136_GM001364	fbpB	1.68	ferric iron ABC transporter permease
Arc9.136_GM002459	ABC.PE.A	1.53	ABC transporter ATP-binding protein
Arc9.136_GM000630	cysA	0.56	sulfate ABC transporter ATP-binding protein
Arc9.136_GM002509	sir	0.62	sulfite reductase
Replication, recombination, and repair			
Arc9.136_GM002892	dnaE2	4.37	error-prone DNA polymerase
Arc9.136_GM002764	DnaB	2.80	Replicative DNA helicase
Arc9.136_GM002761	DNMT1	2.76	DNA cytosine methyltransferase
Arc9.136_GM001599	dinB	3.66	DNA polymerase IV
Arc9.136_GM001729	ABC-2.A	1.83	ABC-2 type transport system ATP-binding protein
Arc9.136_GM001730	ABC-2.P	1.84	ABC-2 type transport system permease protein

## Data Availability

The *Nocardioides arcticus* sp. nov. has been deposited in the China Center for Type Culture Collection Center (CCTCC) with accession number CCTCC AA2023014 and Korean Collection for Type Cultures (KCTC) with accession number KCTC 49996. The complete genome sequence of this strain has been submitted to the GenBank database under accession number CP113431.1.

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
