# Peer review of "Taxonomic Identification of the Arctic Strain Nocardioides Arcticus Sp. Nov. and Global Transcriptomic Analysis in Response to Hydrogen Peroxide Stress"

_ijms, 2023, doi:10.3390/ijms241813943_

Round 1
Reviewer 1 Report
The manuscript presented by the Authors includes research conducted on a new bacterial species acquired during the ninth Arctic scientific expedition. This fact formed the basis for carrying out taxonomic identification of new bacterial species, based on biochemical, physiological properties of the bacteria, as well as chemical composition, among other factors. The result of the authors' research indicated the presence of a new species belonging to the genus Nocardioides, further underscoring the novel nature of the manuscript. I like the way the manuscript is organized, as well as the detail with which the Authors described each step of the study. I only propose to change the order in which the research results are presented to improve clarity. For this purpose, I propose to include "Cell morphology and physiology" as section 2.1. and as section 2.2, I propose to include the results of 16S analysis and preparation of phylogenetic analysis. The manuscript as a whole has been prepared fairly and provides a basis for further research into the antioxidant mechanism of the new species.
Author Response
Comments:
The manuscript presented by the Authors includes research conducted on a new bacterial species acquired during the ninth Arctic scientific expedition. This fact formed the basis for carrying out taxonomic identification of new bacterial species, based on biochemical, physiological properties of the bacteria, as well as chemical composition, among other factors. The result of the authors' research indicated the presence of a new species belonging to the genus Nocardioides, further underscoring the novel nature of the manuscript. I like the way the manuscript is organized, as well as the detail with which the Authors described each step of the study. I only propose to change the order in which the research results are presented to improve clarity. For this purpose, I propose to include "Cell morphology and physiology" as section 2.1. and as section 2.2, I propose to include the results of 16S analysis and preparation of phylogenetic analysis. The manuscript as a whole has been prepared fairly and provides a basis for further research into the antioxidant mechanism of the new species.
Response:
We are grateful to you for your kind comments regarding our research, which give us a great encouragement.
"Cell morphology and physiology" has been revised as section 2.1.
Reviewer 2 Report
Overall evaluation:
The analysis of differentially expressed genes (DEGs) in the manuscript appears to be inadequately executed. The current approach may not provide a comprehensive understanding of the gene networks underlying stress response. Here's what could be improved:
- Enrichment Tests: Utilizing enrichment tests will correct for the background set of genes, thereby reducing bias and providing a more accurate representation of the DEGs. This statistical approach ensures that the identified genes are not merely a result of the overall distribution in the genome but are genuinely associated with the stress response under investigation.
- Use of the STRING Database: Leveraging the STRING database to explore and visualize the gene networks related to stress response would provide valuable insights. This network-based approach could identify key interactions and pathways, enhancing the interpretation of the data.
Discussion Section: The current discussion seems to focus on individual genes without integrating them into a broader context. Utilizing the above-mentioned methods would not only enrich the analysis but also lead to a more cohesive and enlightening discussion. By understanding the interactions and pathways involved, the discussion can transition from isolated gene observations to an integrated interpretation of the stress response mechanism.
Specific comments:
Lines 162-171: It's perplexing that the number of annotated genes is reported as higher than the total number of genes. This discrepancy needs clarification and possible correction. Please ensure that the data presented align with biological understanding.
Figure 2:
A) Please clarify the different tracks represented in the figure. Labels or a legend may be helpful.
B) Consider sorting the barplot by prevalence within each class (i.e., molecular function, cellular component, biological process). Additionally, the
resolution of the figures is too low, making them hard to interpret. Please provide higher-resolution images.
Figure 3:
A) The labels on the tree are uninformative and need to be revised to provide meaningful information. Consider adding clear and descriptive labels that convey the significance or categorization represented in the tree.
B) Sort the categories in descending order by enrichment for easier interpretation.
D) Present the data using only the number of genes, and sort the barplot by descending order. A functional enrichment test based on the union of the expressed genes as a background set would provide more meaningful insights, as some categories might appear prevalent merely due to their abundance in the genome.
Lines 249-260: This section seems more appropriate for the results in a comparative genomic section. Please consider relocating it.
Lines 221-232: This content might fit better in the introduction, as it seems to lay foundational information.
Lines 258-260: Consider moving this to the results and presenting it using a pairwise ANI triangular matrix. The citation of the Nature paper is unclear; please clarify or justify its relevance.
Line 261-262: This line appears to be an incomplete sentence without a verb. Please revise for grammatical correctness.
Lines 281-283: The expectation that superoxide dismutase would not be differentially expressed is unclear. Please provide supporting evidence or rationale.
Discussion Section: The current discussion is somewhat confusing. Consider leveraging the STRING database to analyze the genes. Investigating whether the genes are part of the same network or in a small number of different networks could enhance understanding. Since Nocardioides is available in the STRING database, aligning the genes to one of the reference genomes using tools like Panaroo or other bacterial comparative genomics software may be beneficial.
Reviewer 3 Report
Title “…… Nocardioides arcticensis”
I would like to immediately protest against the authors' attempt to present this work as a description of a "new species". This does not in any way comply with the rules for the description and publication of new taxa of microorganisms. If the authors are ready to return to this issue in the future, I recommend them to visit this resource
https://www.microbiologyresearch.org/content/journal/ijsem?page=about-journal#1
The focus of the work is also unclear to me - what is the main purpose of the study - the taxonomic characteristics of a new strain, or the assessment of the influence of one particular factor on this strain... In combining these two goals, I do not see a deep meaning
Lines 32-92
The introduction is unusually large and does not correspond to the content. You are giving here a part of the scientific review instead of the problems, goals and objectives of this study.
You should carefully read the Rules for Authors, (Research Manuscript Sections): https://www.mdpi.com/journal/ijms/instructions
Line 116-121
I would like to be able to familiarize myself with the 16s rRNA gene sequences. In addition to the fact that the authors indicate that several copies of this gene were found in the genome, there is no information about how different they are and whether this difference affects the configuration of the phylogenetic tree
Line 122-126,
Why didn't the authors also evaluate in silico DDH?
Line 176
It is not entirely clear to me what is the value of finding out the MIC of hydrogen peroxide for this strain? Is there any reason to believe that it has a special stability? Then other microorganisms, both related and more distant, would have to be added to the study...
Line 234 «to grow well on media supplemented with hydrogen peroxide»
Please explain about which media you mean
Lines 238-248
The discussion assumes not just a statement of data, but a comparison and comprehension. The obtained data on the assimilation of substrates and enzymatic activity somehow distinguish this strain from the closest relatives? What was the meaning of setting these tests?
Line 398
«….morphology was described using light microscopy and transmission electron microscopy»
I didn't see any photos or other evidence of using this method in the Results section. Besides. there is no description of the preparation of samples, as well as a link to the methodology.
Lines 398-399
The Gram staining method is of no value from a taxonomic point of view
Lines 406-407
There is neither a description nor references to the methods used to study chemotaxonomic features
Line 480
“Our results describe a new strain of Nocardioides arcticensis, Arc9.136…”
As I understand it, the authors meant that they described a strain -- representing a new species for which they propose a species name. I wrote above that the description of a new species must follow certain rules
Fig.1, A
The authors should also place here the number that was assigned to the 16S gene sequence deposited in the Genbank
Fig.1, B
I would prefer to see the results in the form of a table - simply and clearly
Lines 274-277
Indeed, the data obtained give reason to believe that this is a representative of a new species - however, on the basis of this work, it cannot be included in List of Prokaryotic names with Standing in Nomenclature
Round 2
Reviewer 2 Report
I am pleased to report that the authors have comprehensively addressed all the concerns and suggestions raised in the initial review. I commend the authors for their thorough and thoughtful responses to the review comments. The manuscript is significantly improved and appears to be ready for publication.
Author Response
Dear Reviewer
Thank you so much for your valuable comments which will facilitate the quality of the data analysis and very useful for our other research.
Best wishes,
Bailin Cong
Reviewer 3 Report
The comments are included in attached file

Author Response
Thank you for your comments, please kindly find the attachment of Doc.

Round 3
Reviewer 3 Report
Introduction is overloaded. The authors did not take into account the comments
Build a genome-wide phenotypic tree with a bootstrap of at least 60
